

# Modus operandi and affect in Sweden: the Swedish version of the Regulatory Mode Questionnaire

Clara Amato[1,2,*], Ali Al Nima[1,2], Marko Mihailovic[1,2,3] and Danilo Garcia[1,2,4,5,*]

[1] Blekinge Center of Competence, Blekinge County Council, Karlskrona, Sweden
[2] Network for Empowerment and Well-Being, Sweden
[3] Department of Psychiatry and Behavioral Sciences, Northwestern University, Chicago, IL, United States of America
[4] Department of Psychology, University of Gothenburg, Gothenburg, Sweden
[5] Department of Psychology, Lund University, Lund, Sweden
[*] These authors contributed equally to this work.

Corresponding author
Danilo Garcia,
danilo.garcia@icloud.com

## ABSTRACT

**Background**. The Regulatory Mode Questionnaire (RMQ) is the most used and internationally well-known instrument for the measurement of individual differences in the two self-regulatory modes: locomotion (i.e., the aspect of self-regulation that is concerned with movement from state to state) and assessment (i.e., the comparative aspect of self-regulation). The aim of the present study was to verify the independence of the two regulatory modes, as postulated by the Regulatory Mode Theory (*Kruglanski et al., 2000*), and the psychometric properties of the RMQ in the Swedish context. Furthermore, we investigated the relationship between regulatory modes (locomotion and assessment) and affective well-being (i.e., positive affect and negative affect).

**Method**. A total of 655 university and high school students in the West of Sweden (males = 408 females = 242, and five participants who didn't report their gender; $age_{mean}$ = 21.93 ± 6.51) responded to the RMQ and the Positive Affect Negative Affect Schedule. We conducted two confirmatory factor analyses using structural equation modeling (SEM). A third SEM was conducted to test the relationship between locomotion and assessment to positive affect and negative affect.

**Results**. The first analyses confirmed the unidimensional factor structure of locomotion and assessment and both scales showed good reliability. The assessment scale, however, was modified by dropping item 10 ("I don't spend much time thinking about ways others could improve themselves".) because it showed low loading (.07, $p = .115$). Furthermore, the effect of locomotion on positive affect was stronger than the effect of assessment on positive affect ($Z = -15.16$, $p < .001$), while the effect of assessment on negative affect was stronger than the effect of locomotion on negative affect ($Z = 10.73$, $p < .001$).

**Conclusion** . The factor structure of the Swedish version of the RMQ is, as Regulatory Mode Theory suggests, unidimensional and it showed good reliability. The scales discriminated between the two affective well-being dimensions. We suggest that the Swedish version of the RMQ, with only minor modifications, is a useful instrument to tap individual differences in locomotion and assessment. Hence, the present study contributes to the validation of the RMQ in the Swedish culture and adds support to the theoretical framework of self-regulatory mode.

## INTRODUCTION

The Regulatory Mode Questionnaire (RMQ; *Kruglanski et al., 2000*) measures individual differences in the way people prefer to operate when striving to reach a goal. Specifically, the RMQ measures how individuals differ in their tendency to move from state to state towards a goal (locomotion) and in their tendency to compare different alternatives, such as, the value of different goals or the different means to reach a chosen goal (assessment). Structurally, the RMQ is constituted by two scales that tap individual differences in locomotion and assessment.[1] So far, the RMQ has been translated into several languages (e.g., French, Italian, Spanish, Japanese, Hebrew, Hindu, and Korean), including Swedish (see Table 1; *Jimmefors et al., 2014*; *Garcia et al., 2015c*; *Garcia & Archer, 2016*; *Garcia & Lindskär, 2016*; *Garcia et al., 2017e*). A few studies, have demonstrated the unidimensionality, the internal consistency, and the temporal stability as well as the convergent and discriminant validity of each of the two scales in the RMQ (*Kruglanski et al., 2000*). However, with exception of the Italian and the original English version, the structure and psychometric properties of these versions of the RMQ have not yet been investigated. In the present study, we test the unidimensionality and reliability of the Swedish version of the RMQ scales (locomotion and assessment). Furthermore, we also investigate the relationship between regulatory modes and affective well-being (i.e., positive affect and negative affect) in the Swedish sample at hand. Importantly, positive affect and negative affect are not only indicators or markers of well-being (*Diener, 1984*), these two affectivity dimensions are two distinctive factors that reflect stable emotional-temperamental dispositions and form together an adaptive complex signal sensitivity system (e.g., *Watson & Clark, 1994*; *Watson, Clark & Tellegen, 1988*; *Tellegen, 1993*; *Garcia, 2011*; *Garcia, in press-a*). In the following sections, we will introduce Regulatory Mode Theory (*Kruglanski et al., 2000*), we will provide an overview of the wide range of use of the RMQ, and we will develop the expected associations between the regulatory modes (locomotion and assessment) and the affective well-being dimensions (i.e., positive and negative affect).

### Regulatory Mode Theory: locomotion and assessment orientations

In the classic theory of control (*Carver & Scheier, 1990*), assessment and locomotion are functionally interdependent parts of a single self-regulatory process. In the first step, through the assessment function, an individual compares the different mental goal representations and chooses the favorite one, then he/she compares the current state with the desired end-state. In the second step, through the locomotion function, the individual tries to reduce the current-end states discrepancy by starting and maintaining movement toward the target goal. Conversely, in the frame of Regulatory Mode Theory (*Higgins, Kruglanski & Pierro, 2003*), locomotion and assessment are independent functions, which can receive different emphasis. In this context, locomotion is the aspect of self-regulation concerned with ''movement from state to state and with committing the psychological

[1] Although the RMQ also contains a lie scale, consisting of six items (see white cells in Table 1), this scale is often not used in most research and some studies have also applied only one of the scales (e.g., locomotion; *Pierro et al., 2013*).

**Table 1** The Swedish translation of the Regulatory mode questionnaire.

| Item no. | English | Swedish | Keywords |
|---|---|---|---|
| 1. | I don't mind doing things even if they involve extra effort. | Jag har inget emot att göra saker även om det innebär en extra ansträngning. | Extra effort |
| 2(R). | I never evaluate my social interactions with others after they occur. | Jag utvärderar aldrig mina sociala samspel med andra efter att de hänt. | Evaluate interactions |
| 3. | I am a "workaholic". | Jag är en "arbetsnarkoman". | Workaholic |
| 4. | I feel excited just before I am about to reach a goal. | Jag känner mig upprymd precis innan jag uppnår ett mål. | Excited |
| 5. | I enjoy actively doing things, more than just watching and observing. | Jag tycker bättre om att aktivt göra saker istället för att bara titta på och observera. | Actively do things |
| 6. | I spend a great deal of time taking inventory of my positive and negative characteristics. | Jag lägger ner mycket tid på att inventera mina positiva och negativa karaktärsdrag. | Inventory of characteristics |
| 7. | I like evaluating other people's plans. | Jag tycker om att utvärdera andra människors planer. | Evaluate others' plans |
| 8. | I am a "doer''. | Jag är en människa som får saker och ting gjorda. | Doer |
| 9. | I often compare myself with other people. | Jag jämför ofta mig själv med andra. | Compare to others |
| 10(R). | I don't spend much time thinking about ways others could improve themselves. | Jag spenderar inte mycket tid på att tänka på hur andra kan förbättra sig själva. | Not improve others |
| 11. | I often critique work done by myself and others. | Jag kritiserar ofta mitt och andras arbete. | Critique self and others |
| 12. | I believe one should never engage in leisure activities. | Jag anser att man aldrig bör engagera sig i fritidsaktiviteter. | |
| 13(R). | When I finish one project, I often wait awhile before getting started on a new one. | När jag avslutar ett projekt, väntar jag ofta ett tag innan jag påbörjar nästa. | Wait before start |
| 14. | I have never been late for work or for an appointment. | Jag har aldrig varit sen till arbetet eller ett avtalat möte. | |
| 15. | I often feel that I am being evaluated by others. | Jag känner ofta att jag blir bedömd av andra. | Evaluated by others |
| 16. | When I decide to do something, I can't wait to get started. | När jag bestämt mig för att göra något kan jag inte sätta igång snabbt nog. | Can't wait |
| 17. | I always make the right decision. | Jag fattar alltid rätt beslut. | |
| 18. | I never find faults with someone I like. | Jag hittar aldrig några fel på människor jag tycker om. | |
| 19. | I am a critical person. | Jag är en kritisk person. | Critical person |
| 20. | I am very self-critical and self-conscious about what I am saying. | Jag är väldigt självkritisk och självmedveten om vad jag säger. | Self-critical and self-conscious |
| 21. | By the time I accomplish a task, I already have the next one in mind. | Medans jag slutför ett projekt har jag redan börjat tänka på nästa. | Next task in mind |
| 22. | I often think that other people's choices and decisions are wrong. | Jag tycker ofta att andra människors val och beslut är fel. | Others are wrong |

**Table 1** (*continued*)

| Item no. | English | Swedish | Keywords |
|---|---|---|---|
| 23. | I have never hurt another person's feelings. | Jag har aldrig sårat en annan människas känslor. | |
| 24(R). | I am a "low energy" person. | Jag är en människa med låg energi. | Low energy |
| 25. | Most of the time my thoughts are occupied with the task that I wish to accomplish | För det mesta kretsar mina tankar kring den uppgift som jag vill slutföra. | Thoughts occupied with tasks |
| 26. | I feel that there is no such thing as an honest mistake. | Jag anser att det inte finns några ärliga misstag. | |
| 27(R). | I rarely analyze the conversations I have had with others after they occur. | Jag analyserar sällan konversationer jag haft med andra efter att de inträffat. | Rarely analyze conversations |
| 28. | When I get started on something, I usually persevere until I finish. | När jag påbörjar något håller jag allt som oftast ut tills jag slutfört det. | Persevere until finish |
| 29. | I am a "go-getter". | Jag är en handlingsmänniska. | Go-getter |
| 30. | When I meet a new person I usually evaluate how well he or she is doing on various dimensions (e.g., looks, achievements, social status, clothes). | När jag träffar en ny människa bedömer jag oftast hur han eller hon klarar sig på olika plan (t. ex utseende, prestationer, social status, kläder). | Evaluate dimensions |

**Notes.**
Light grey cells, locomotion items; dark grey cells, assessment items; white cells, lie scale (not used in the present study); and (R), reversed item. Item 10 was removed from the final analysis due to low loading. Adapted from *Garcia et al. (2017e)*, with permission from D Garcia.

resources that will initiate and maintain goal-related movement in a straightforward and direct manner, without undue distractions or delays'' (*Kruglanski et al., 2000*, p. 794) and assessment is ''the comparative aspect of self-regulation concerned with critically evaluating entities or states, such as goals or means, in relation to alternatives in order to judge relative quality'' (*Kruglanski et al., 2000*, p. 794).

There are two important assumptions in Regulatory Mode Theory: (1) temperament and socialization lead individuals to develop different degrees of concern with *movement* or progress (i.e., emphasizing locomotion) and different degrees of concern for standards and for *critically evaluating* alternatives (i.e., emphasizing assessment) and (2) the reasons for preferring locomotion or assessment are unrelated (*Kruglanski et al., 2000*). In other words, locomotion and assessment are independent and orthogonal biopsychosocial modes of operation, thus, some individuals can be high on one mode and low on the other mode, low on both modes, or high on both modes (cf. *Garcia & Lindskär, 2016*). The RMQ is indeed an instrument that helps to measure the chronic preference for one regulatory mode orientation over the other.

## Applications of the RMQ

Previous studies have shown the high practical relevance and the wide potential applicability of the RMQ. Since its development, several studies have shown that the two regulatory modes are related with many important aspects, ranging from organizational behavior (e.g., preference for leadership style; *Benjamin & Flynn, 2006*; time management; *Amato et al., 2014*), to entrepreneurial aptitude (*Amato et al., 2016*), to cognition (e.g., cognitive biases, counterfactual thinking; *Pierro et al., 2008*), and economic decision-making (e.g.,

inter-temporal choices; *Mannetti et al., 2009*). Particularly relevant to the present study is the strong link between self-regulation and both health and well-being. Specifically, there is evidence that a prevalence of locomotion orientation is positively related to affective well-being (i.e., frequent experience of positive affect and infrequent experience of negative affect), while a prevalence of assessment orientation is associated with maladaptive psychological functioning and even psychopathological symptoms (*Garcia et al., 2016b*). For instance, locomotion is both indirectly and directly positively related to affective well-being. For example, people who are high in locomotion do not engage in social comparison and have therefore a tendency to be less materialistic, which in turn leads them to experience less negative affect (*Giacomantonio, Mannetti & Pierro, 2013*). Indeed, engaging in social comparison and self-evaluation, something a person does when in assessment mode, leads to materialistic thinking, which in turn leads to high levels of negative affect (*Giacomantonio, Mannetti & Pierro, 2013*). That being said, affective well-being is more that the absence of negative emotions (cf. *Cloninger, 2004*). In this context, people who are high in locomotion report having a sense of life purpose and this lead them to be more satisfied with their life (*Vazeou-Nieuwenhuis, Orehek & Scheier, 2017*). In addition, locomotion is directly related to higher levels of vitality (*Kruglanski et al., 2000*), higher levels of optimism (*Kruglanski et al., 2000*), happiness (*Smith et al., 1993*) and last but not the least, positive affect (*Kruglanski et al., 2000*; *Garcia et al., 2015c*; *Garcia et al., 2016b*). Individuals with a high assessment orientation, on the other hand, are more likely to present depressive symptoms (*Hong, Tan & Chang, 2004*), they are also more susceptible to suffer from borderline disorder (*Bornovalova et al., 2008*) and obsesive-compulsive disorder (*Shalev & Sulkowski, 2009*), and report high levels of negative affect (*Kruglanski et al., 2000*; *Garcia et al., 2015c*; *Garcia et al., 2016b*).

## The present study

In the present study, based on Regulatory Mode Theory, we test the unidimensionality of the Swedish version of the RMQ scales (locomotion and assessment). We also test the reliability of the Swedish RMQ and the predictive validity of the regulatory mode scales by investigating the relationship between regulatory modes (locomotion and assessment) and affective well-being (positive and negative affect). The affectivity dimensions to test the RMQ scales' predictive validity were found to be appropriate for different reasons.

First of all, the affectivity dimension range from pleasant/unpleasant engagement (for example: enthusiastic and active for positive affect, anger and fear for negative affect), to unpleasant/pleasant disengagement (for example: sad and bored for positive affect, calm and serene for negative affect) (*Watson, Clark & Tellegen, 1988*; *Watson & Clark, 1994*). Thus, as locomotion and assessment (*Kruglanski et al., 2000*), positive affect and negative affect are best conceptualized as two independent orthogonal dimensions that are part of a complex adaptive system and not as ends of one single dimension (*Garcia, 2011*; *Garcia, in press-a*). These two affectivity dimensions reflect approaching behavior (positive affect) and avoidant behavior (negative affect) (*Garcia, 2011*; *Garcia, in press-a*). Thus, we expected locomotion to predict positive affect or approaching behavior and assessment to predict negative affect or avoidant behavior (cf. with *Kruglanski et al., 2000*, who also suggested a

positive correlation between locomotion and positive affect because locomotion or forward movement contributes to a sense of progress and a positive correlation between assessment and negative affect because high assessment involves ruminating self-evaluation that may highlight the discrepancies between one's actual self and a desired self).

Secondly, according to Regulatory Mode Theory, temperament and socialization lead individuals to develop different degrees of locomotion and assessment. Indeed, Kruglanski and colleagues *(2000)* found a positive correlation between locomotion and extraversion ($r = .38$; $p < .001$), a negative correlation between locomotion and neuroticism ($r = -.20$; $p < .001$), a positive correlation between assessment and neuroticism ($r = .41$; $p < .001$), and a negative correlation between assessment and extraversion ($r = -.03$; *nonsignificant*). Notably, positive and negative affect are almost identical with these personality traits (i.e., extraversion and neuroticism, respectively) (*Larsen & Ketelaar, 1991*; see also *Garcia et al., 2015a*; *Garcia et al., 2015b*; *Rapp Ricciardi et al., 2014*; *Garcia & Moradi, 2013*; *Garcia, 2012*; *Garcia et al., 2010*). In addition, compared to extraversion and neuroticism, the two affectivity dimensions involve more mood and social traits (*Gunderson et al., 1999*). Therefore, these affectivity dimension are not only seen as temperamental dispositions or signal sensitivity systems, but also as being complementary to extraversion and neuroticism (*Tellegen, 1993*). We argue that this supports our expectation of a positive relationship between locomotion and positive affect and between assessment and negative affect. In addition, this suggests a negative or non-significant relationship between locomotion and negative affect and between assessment and positive affect. Hence, we expected a stronger relationship between locomotion and positive affect than between assessment and positive affect. Conversely, we expected a stronger relationship between assessment and negative affect than between locomotion and negative affect.

## MATERIALS AND METHODS

### Ethical statement

After consulting with the Network for Empowerment and Well-Being's Review Board we arrived at the conclusion that the design of the present study (e.g., all participants' data were anonymous and will not be used for commercial or other non-scientific purposes) required only informed consent from the participants.

### Participants and procedure

A total of 655 students (357 university students and 298 high school students) in the West of Sweden (males = 408 females = 242, and five participants who didn't report their gender, age$_{mean}$ = 21.93 ± 6.51) participated in the study. All participants were informed that taking part of the study was voluntary and anonymous. They were presented with paper-pencil Swedish versions of the RMQ and the Positive Affect Negative Affect Schedule.

### Measures

*Self-regulation.* The Swedish version (*Garcia et al., 2015c*; *Garcia et al., 2017e*) of the RMQ (*Kruglanski et al., 2000*) was used to assess self-regulatory mode/orientations. The test consists of 30 items measured on a 6-point likert scale (from 1 = *strongly disagree* to

**Table 2** Correlations, *Cronbach's alpha, Means* and *SD* (±) for all variables in the study.

| | | Affective well-being | | Regulatory mode | |
| --- | --- | --- | --- | --- | --- |
| | | Positive affect | Negative affect | Locomotion | Assessment |
| Affective well-being | Positive affect | – | | | |
| | Negative affect | .29 [**] | – | | |
| Self-regulatory mode | Locomotion | .53 [**] | −.13 [**] | – | |
| | Assessment | −.13 [**] | .40 [**] | .08 [*] | – |
| | *Mean* and *SD* (±) | 3.54 ± 0.64 | 2.21 ± 0.67 | 3.96 ± 0.66 | 3.53 ± 0.69 |
| | *Cronbach's alpha* | .85 | .90 | .76 | .79 |

**Notes.**
[*] $p < .05$.
[**] $p < .01$.
Light grey cells, significant correlations between participants' locomotion and affectivity scores; dark grey cells, significant correlations between participants' assessment and affectivity scores; black cells, significant correlations within self-regulatory modes and affectivity dimensions, respectively.

6 = *strongly agree*); 12 items measuring assessment (e.g., "I spend a great deal of time taking inventory of my positive and negative characteristics", "I am a critical person"), 12 items measuring locomotion ("I am a doer", "When I get started on something, I usually persevere until I finish it"), and six constituting a lie scale that was not analyzed in the present study (see Table 1).

*Affect.* The Positive Affect and Negative Affect Schedule (*Watson, Clark & Tellegen, 1988*) assesses the affective component of subjective well-being by requiring participants to rate to what extent (1 = *very slightly*, 5 = *extremely*) during the last few weeks they have experienced 10 positive (e.g., strong, proud) and interested and 10 negative (e.g., afraid, ashamed, and nervous) affective states. The Swedish version has been used in previous studies with good psychometric properties (e.g., *Schütz, Archer & Garcia, 2013*).

## Preliminary descriptive analyses

We conducted Preliminary analyses to test some statistical issues, such as the presence of outliers and collinearity between variables. We first standardized the scores of each variable and tested if any cases had larger standardized scores than ±3.29, as recommended by *Tabachnick & Fidell (2007)*. The analysis detected no cases of univariate outliers. The variables had skewness between −.10 to .64 and kurtosis between .70 and .12, thus, indicating that the data did not violate the assumption of normality (see *Tabachnick & Fidell, 2007*, p. 80). Moreover, we examined the distributions of the items and linearity by scatter plots. The results showed the absence of any signs of violation of normality and linearity assumptions. Finally, all the significant correlations among the variables were not so high as to threaten statistical multicollinearity and singularity. Indeed, the correlation coefficients range between .08 and .53, thus not constituting a cause for multicollianiarity and singularity issues (see, *Tabachnick & Fidell, 2007*, p. 88). Table 2 show the correlations between variables in the presents study, along Cronbach's alphas and both means and standard deviations for each variable.

## RESULTS

Based on Regulatory Mode Theory, we first tested the unidimensionality of the Swedish version of the RMQ scales (locomotion and assessment) using two separate confirmatory factor analyses through structural equation modeling (SEM), one for locomotion and one for assessment. We used separate analyses since locomotion and assessment are two independent unrelated modes of operation (e.g., *Kruglanski et al., 2000*).

### Locomotion

The first confirmatory analysis was conducted using the 12 items to estimate locomotion as one factor. The analysis showed that *chi-square* value was significant ($Chi^2 = 268.79$, $df = 54$, $p < .001$), the *goodness of fit index* was .93 and *the root mean square error of approximation* for the default model was .08. The *chi-square* statistic is heavily influenced by sample size (*Kline, 2010*), with larger samples (in this study $N = 650$) leading to larger value and therefore, a larger likelihood of being significant. However, other values (*the goodness of fit index* and *the root mean square error of approximation*) indicated that the hypothesized model fits the data well. All the regression weights/loadings between the factor (i.e., locomotion) and its 12 items were significant at $p < .001$ (ranging from .32 to .72) with the exception of two items that had low loadings (.29 for the reversed item 13: "When I finish one project, I often wait awhile before getting started on a new one." and .15 for item 16: "When I decide to do something, I can't wait to get started".) which were also significant at $p < .001$ (see Fig. 1 and Table 3). Nevertheless, the results mirror those in the original article (*Kruglanski et al., 2000*) in which the fit of the one-factor model for both the locomotion and assessment scales were higher than .90 for *goodness of fit index*, RMRs around .10, and the overall *alpha* for locomotion was .82.

### Assessment

The second confirmatory analysis was conducted using the 12 items to estimate assessment as one factor. The analysis showed that *chi-square* value was significant ($Chi^2 = 362.20$, $df = 54$, $p < .001$), the *goodness of fit index* was .91 and *the root mean square error of approximation* for the default model was .09. However, based on other important statistical indexes, such as *the goodness of fit index and the root mean square error of approximation*, we assumed that the model fits the data well. All the regression weights/loadings between the factor (i.e., assessment) and its items were significant at $p < .001$ (ranging from .26 to .66) with the exception of reversed item 10 ("I don't spend much time thinking about ways others could improve themselves".) which had low loading .07 (*ns*, $p = .115$) (see Fig. 2). Hence, item 10 was removed from the analyses. After this, the analysis showed that *chi-square* value was significant ($Chi^2 = 336.88$, $df = 44$, $p < .001$), the *goodness of fit index* was .91 and *the root mean square error of approximation* for the model was .10 (see Fig. 3 and Table 3). All the regression weights/loadings between the factor (i.e., assessment) and its 11 items were significant at $p < .001$ with ranging from .26 to .66. Thus, the model with just 11 items fits the data well.

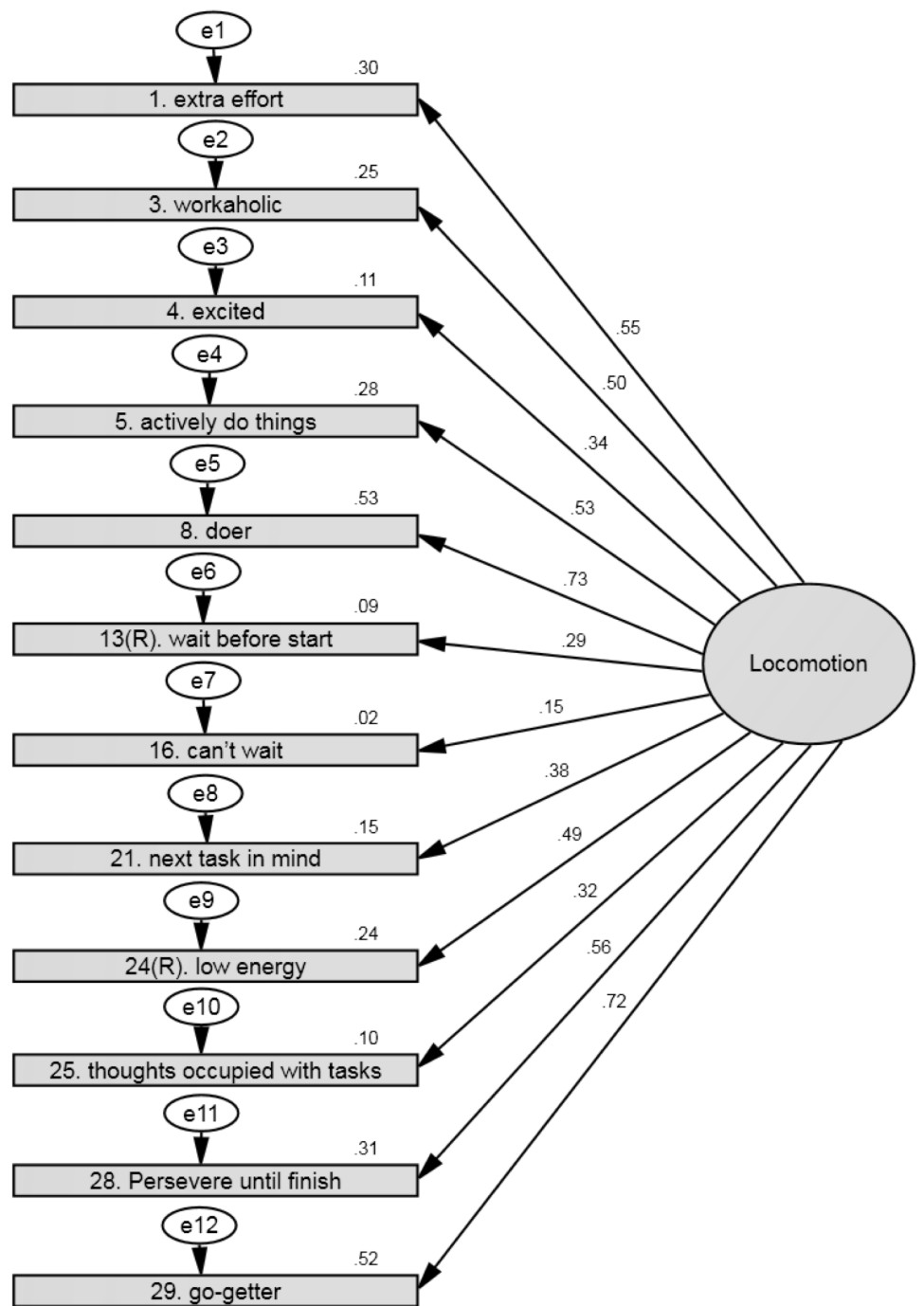

**Figure 1** **SEM showing the standardized parameter estimates between the latent factor (i.e., locomotion) and the 12 locomotion items from the RMQ.** Note: *Chi square* = 268.79, *df* = 54, *p* < .001; *goodness of fit index* = .93 and the *root mean square error of approximation* = .08 (*N* = 650).

**Table 3** Standardized and unstandardized structural coefficients for the regression weights/loadings between the Swedish version of the RMQ-scales locomotion and assessment and its respective items ($N = 650$).

| Regulatory mode | Item | β | SE | B | P |
|---|---|---|---|---|---|
| Locomotion | 1. Extra effort. | .55 | .11 | 1.00 | <.001 |
| | 3. Workaholic | .50 | .11 | 1.07 | <.001 |
| | 4. Excited | .34 | .08 | 0.61 | <.001 |
| | 5. Actively doing things | .53 | .09 | 0.96 | <.001 |
| | 8. Doer | .73 | .11 | 1.31 | <.001 |
| | 13(R). Wait before start | .29 | .10 | 0.62 | <.001 |
| | 16. Can't wait | .15 | .09 | 0.29 | <.001 |
| | 21. Next task in mind | .38 | .10 | 0.82 | <.001 |
| | 24(R). Low energy | .49 | .11 | 1.08 | <.001 |
| | 25. Thoughts occupied with task | .32 | .09 | 0.60 | <.001 |
| | 28. Persevere until finish | .56 | .11 | 1.13 | <.001 |
| | 29. Go-getter | .72 | .10 | 1.29 | <.001 |
| Assessment | 2(R). Evaluate interactions | .26 | .10 | 0.54 | <.001 |
| | 6. Inventory of characteristics | .46 | .12 | 0.99 | <.001 |
| | 7. Evaluate other's plans | .53 | .13 | 1.16 | <.001 |
| | 9. Compare to others | .55 | .14 | 1.21 | <.001 |
| | 11. Critique self and others | .63 | .14 | 1.31 | <.001 |
| | 15. Evaluated by others | .49 | .12 | 1.00 | <.001 |
| | 19. Critical person | .66 | .14 | 1.38 | <.001 |
| | 20. Self-critical and self-conscious | .64 | .13 | 1.23 | <.001 |
| | 22. Others are wrong | .48 | .11 | 0.92 | <.001 |
| | 27(R). Rarely analyze conversations | .42 | .12 | 0.91 | <.001 |
| | 30. Evaluate dimensions | .44 | .11 | 1.00 | <.001 |

## Locomotion, assessment, and affective well-being

In the third SEM, we conducted a path analysis using locomotion (comprising 12 items) and assessment (comprising 11 items) as the independent variables and positive affect and negative affect as the dependent variables. The analysis showed that *chi-square* value was significant (Chi$^2$ = 27.85, $df = 1$, $p < .001$). However, as for the other two models previously presented, the *goodness of fit index* (.98), *comparative fit index* (.93), *incremental fit index* (.93), and *normed fit index* (.93) indicated that the hypothesized model fits the data well. All the regression weights between the independent variables (locomotion, assessment) and the dependent variables (positive affect and negative affect) were significant at $p < .001$. Standardized parameter estimates of correlation and direct effects appear in Fig. 4 and both standardized and unstandardized coefficients are listed in Table 4.

Additionally, we tested assessment and locomotion's equalities of regression paths by comparing the regression weights between locomotion-positive affect to assessment-postive affect and between locomotion-negative affect and assessment-negative affect. The result showed that the effect of locomotion on positive affect was stronger than the effect of assessment on positive affect (Critical Ratios for Differences between Parameters/*Z*

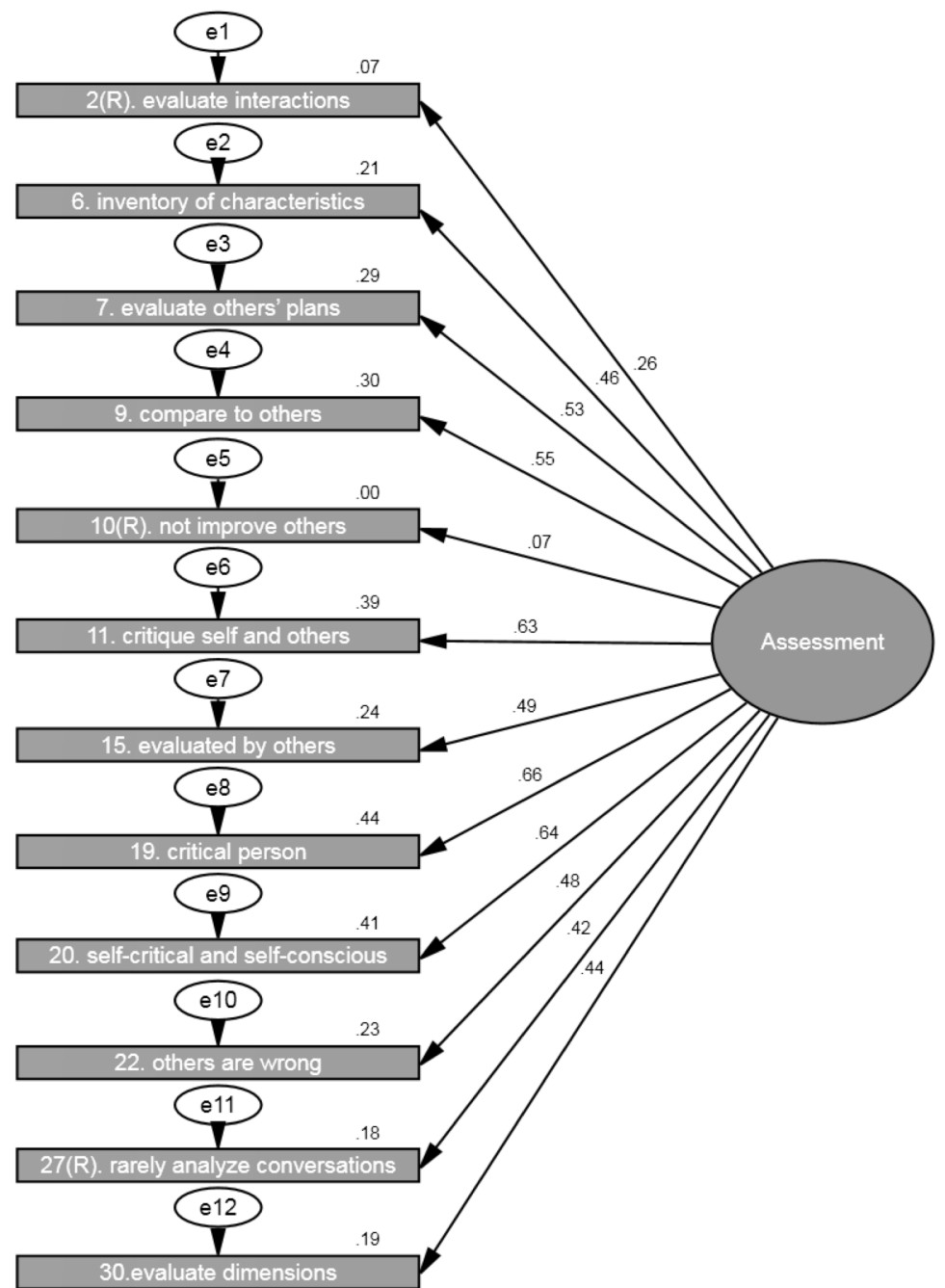

**Figure 2** SEM showing the standardized parameter estimates between the latent factor (i.e., assessment) and the 12 assessment items from the RMQ. Note: *Chi square* = 362.20, *df* = 54, *p* < .001; *goodness of fit index* = .91 and the *root mean square error of approximation* = .09 (*N* = 650).

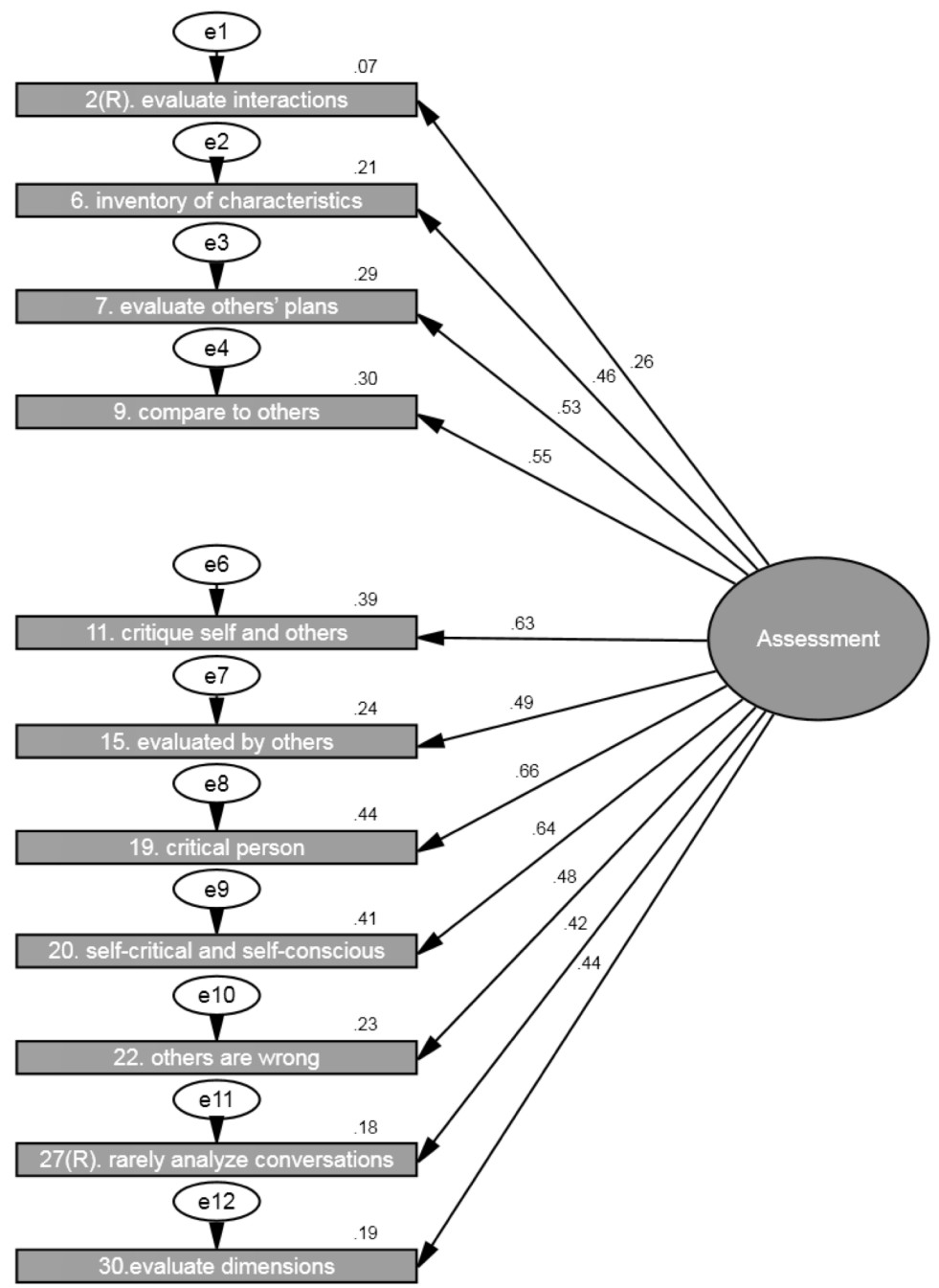

**Figure 3** **SEM showing the standardized parameter estimates among the latent factor (i.e., assessment) and the 11 assessment items (i.e., without item 10) from the RMQ.** Note: *Chi square* = 336.88, *df* = 44, *p* < .001; *goodness of fit index* = .91 and the *root mean square error of approximation* = .10 (*N* = 650).

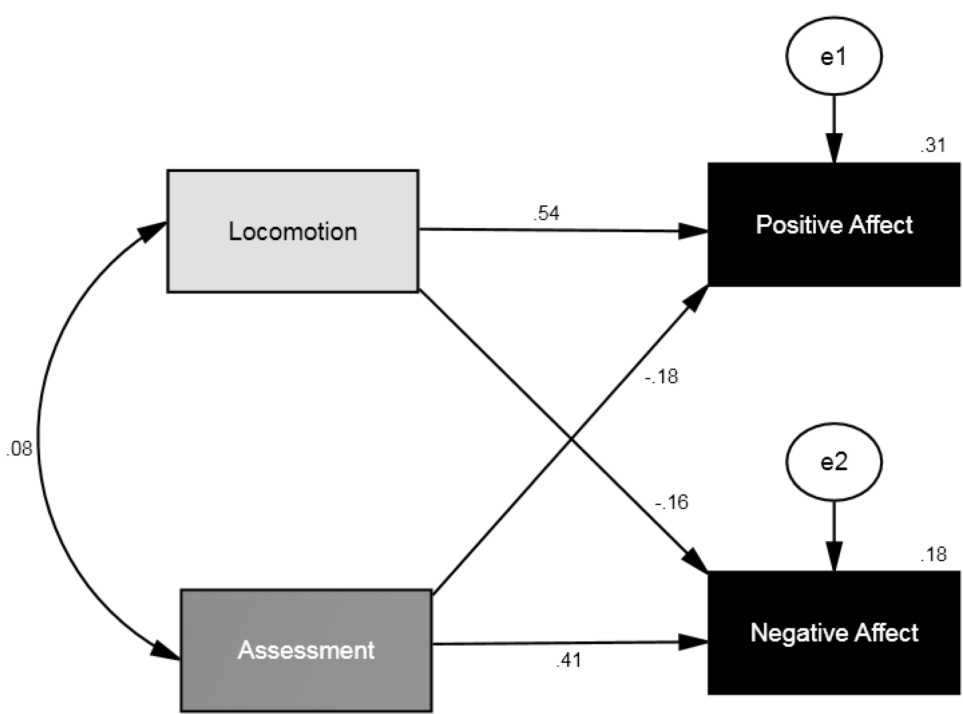

**Figure 4  SEM showing the standardized parameter estimates for the relationship between self-regulatory modes (locomotion and assessment) and affective well-being (positive affect and negative affect).** Note: *Chi square* = 27.85, *df* = 1, *p* < .001; *goodness of fit index* was .98, *comparative fit index* = .93, *incremental fit index* = .93 and *normed fit index* = .93 (*N* = 650).

**Table 4  Standardized and unstandardized structural coefficients for the relationship between self-regulatory modes (locomotion and assessment) and affective well-being (positive affect and negative affect) (*N* = 650).**

| Predictor | Outcome | $\beta$ | SE | $B$ | $p$ |
|---|---|---|---|---|---|
| Locomotion | Positive affect | .54 | .03 | .53 | <.001 |
| Assessment |  | −.18 | .03 | −.16 | <.001 |
| $R^2$ | .31 |  |  |  |  |
| Locomotion | Negative affect | −.16 | .04 | −.16 | <.001 |
| Assessment |  | .41 | .03 | .39 | <.001 |
| $R^2$ | .18 |  |  |  |  |

**Notes.**

Light grey cells, the effect of locomotion on positive affect was stronger than the effect of assessment on positive affect ($Z$ = −15.16, $p$ < .001); dark grey cells, the effect of assessment on negative affect was stronger than the effect of locomotion on negative affect ($Z$ = 10.73, $p$ < .001).

statistic $= -15.16$, $p < .001$), while the effect of assessment on negative affect was stronger than the effect of locomotion on negative affect (Critical Ratios for Differences between Parameters/$Z$ statistic $= 10.73$, $p < .001$). See Table 3.

## DISCUSSION

Over 10 years of research on self-regulation show that individual differences in regulatory mode tendencies affect many aspects of human behavior, and, consequently, health and well-being. The aim of the present study was twofold: (1) to test the applicability of the RMQ in the Swedish context by examining the unidimensionality of the two RMQ scales (i.e., locomotion scale and assessment scale) and their reliability; (2) to test the predictive and discriminant validity of the locomotion and assessment scales by investigating the relationship between the two regulatory modes (i.e., locomotion and assessment) and the affective well-being dimensions (i.e., positive affect and negative affect). First of all, as far as we know, the theorized unidemensionallity of the two modes of regulation has been empirically tested only in the English and the Italian versions of the RMQ (*Kruglanski et al., 2000*). Thereby the importance of providing further support to the validity of RMQ across different cultures. Secondly, to our understanding,[2] the regulatory modes are theorized as biopsychosocial in nature (i.e., influenced by temperament, psychological resources, and socialization), thus, positive affect and negative affect were found appropriate for predictive and discriminant validity analyses. After all, besides being markers of well-being, positive and negative affect are independent and stable temperamental dispositions that involve more mood and social traits than extraversion and neuroticism (e.g., *Watson & Clark, 1994*; *Watson, Clark & Tellegen, 1988*; *Tellegen, 1993*; *Garcia, 2011*; *Garcia, in press-a*).

At a general level, the results obtained here show that both locomotion and assessment have a unidimensional factor structure, thus confirming the findings of previous research conducted with other samples (*Kruglanski et al., 2000*). The Swedish RMQ revealed to also have good indexes of reliability for both scales. That being said, the locomotion reversed item 13 ("When I finish one project, I often wait awhile before getting started on a new one".) and item 16 ("When I decide to do something, I can't wait to get started".) showed significant but relatively low loadings (.29 and .15, respectively). In addition, the assessment reversed item 10 ("I don't spend much time thinking about ways others could improve themselves".) showed low loading (.07, $p = .115$) and was discarded from the final analyses. With regard to the reversed items in both scales (items 10 and 13), despite the potential benefits (e.g., minimizing the problems of inattention and acquiescence), one of the problems derived from using reversed items is that they indeed produce unexpected factor structures (e.g., *Netemeyer, Bearden & Sharma, 2003*) and miscomprehension (*Swain, Weathers & Niedrich, 2008*). These problems are actually more marked among subcultural groups, such as, ethnic and racial minorities (*Steenkamp & Burgees, 2002*). In fact, these problems seem to be more pronounced when scales are used in cultures that differ in values, customs, and language from the original culture in which the scale was developed (cf. *Wong, Rindfleisch & Burroughs, 2003*). For instance, Sweden has the fourth place and Italy, one of the few countries in which the RMQ has been

[2]To the best of our knowledge this is the first scientific article referring to the regulatory modes as a biopsychosocial complex adaptive system (cf. *Garcia, in press-a*). Nevertheless, others (*Garcia & Lindskär, 2016*) have assumed this complex interaction by suggesting regulatory mode profiles: assessor (i.e., high in assessment/low in locomotion), low regulator (i.e., low assessment/low locomotion), high regulator (i.e., high assessment/high locomotion), and locomotor (i.e., low assessment/high locomotion).

[3]A description of a pattern of group behavior towards individuals within Nordic countries that negatively portrays and criticizes individual success and achievement as unworthy and inappropriate. Used generally in colloquial speech in the Nordic countries as a sociological term to describe a condescending attitude towards individuality and success, the term refers to a mentality that de-emphasizes individual effort and places all emphasis on the collective, while simultaneously denigrating those who try to stand out as individual achievers. Retrieved from https://en.wikipedia.org/wiki/Law_of_ Jante#cite_note-3.

validated, the fiftieth place in the 2016 report from the World Economic Forum's Global Gender Gap (see https://www.weforum.org/reports/the-global-gender-gap-report-2016). Additionally, Sweden is an individualistic culture (*Kjell et al., 2013*) and widely known for Jantelagen,[3] suggesting that Swedes have a tendency to not show off by presenting themselves as better than someone else (*Fayolle, Kyrö & Ulijin, 2005*). Probably explaining why, a question with a statement such as: "I don't spend much time thinking about ways others could improve themselves." (assessment reversed item 10), is problematic in the Swedish context. One way or another, we strongly recommend the replication of the present study in other Swedish samples before totally removing this specific item from future studies.

Furthermore, locomotion predicts high (*vs.* low) positive affect, while assessment predicts high (*vs.* low) negative affect. Locomotion orientation is by definition the aspect of self-regulation concerned with maintaining goal-directed progress in a straightforward manner, without undue distractions or delays (*Kruglanski et al., 2000*); individuals with high locomotion orientation have a high behavioral activation system (*Higgins, Kruglanski & Pierro, 2003*), they engage in goal pursuit promptly and exhibit a stronger task orientation (i.e., the tendency to attend to an activity and persist conscientiously until completion). Furthermore, individuals high in locomotion manage their time more efficiently and, consequently, they feel to have control over the time (*Amato et al., 2014*), which is a very important resource for them (*Amato et al., 2017*). These features result in a sense of self-regulatory competence and self-maintenance (i.e., maintaining control over attention) among individuals high in locomotion. Indeed, individuals high on positive effect show these features as well (*Cloninger & Garcia, 2015*). Within this picture, the results of the present study about the relationship between locomotion and affective well-being, are completely consistent with previous findings describing individuals with high locomotion orientation as *active people*, *full of energy, with full concentration* (i.e., high Positive affect; *Watson, Clark & Tellegen, 1988*). At the same time, the tendency to move forward, typical of the locomotion orientation, lead individuals high in locomotion to pass away from negative feelings (e.g., guilt; regret; *Pierro et al., 2008*; A Pierro et al., 2014, unpublished data) and *to not stay engaged with unpleasant mode states* (i.e., low Negative affect, *Watson, Clark & Tellegen, 1988*). On the other hand, the tendency to make social comparisons and to engage frequently in self-evaluation, typical of the assessment orientation, may lead individuals high in assessment to struggle with a large discrepancy perceived between current state and external standards, and between real attainments and the ideal end-states. This determine a subjective distress characterized by the presence of several aversive mood states among individuals high in assessment (i.e., high negative affect). These findings confirm the key role of regulatory mode in affective well-being (*Garcia & Lindskär, 2016*) and confirm the discriminant and predictive validity of the RMQ in the Swedish context.

## Limitations and suggestions for future research

To provide a valid instrument to use in the Swedish context is useful at the theoretical, research, and practical levels. At a theoretical level, the results presented here support the notion of two independent constructs of self-regulation that lead to different approach

(positive affect) and prevention (negative affect) behaviors (cf. *Garcia, 2011*; *Garcia, in press-a*). That being said, at this point a fully biopsychosocial theory of self-regulation needs to be developed using holistic models, such as, Cloninger's ternary model of human personality (*Cloninger, 2004*). This model disentangles personality in biological (harm avoidance, novelty seeking, reward dependence, and persistence), psychological (self-directedness and cooperativeness), and social (self-transcendence) aspects of personality (*Cloninger, 2004*; *Garcia et al., 2017c*; *Garcia et al., 2017a*; *Garcia et al., 2017b*; *Garcia et al., 2017d*). Hence, we suggest that future research should find the Cloninger personality model useful to test if self-regulatory mode theory is biopsychosocial in nature. As far as we understand the theory, self-regulation should be biopsychosocial; but no other work has addressed the concept holistically. At the research level, the Swedish RMQ could be used in the fields of motivation, health, personality, and organizational settings.

At the practical level, the RMQ could be a useful tool of evaluation of current social challenges in Sweden. For instance, Swedish students report grades that are among the lowest of the other 32 OECD countries (PISA, *Sverigesradio, 2013*; *SNAE, 2009*; *SNAE, 2012*) along deteriorations in psychological health (*Fleming et al., 2013*; *Ghofranipour et al., 2013*). At the same time, many Swedish high schools decrease the number of physical education hours available for their students (*Sverigesradio, 2013*)—a type of activity that is well-documented for positive effects on psychological health and well-being among adolescents and young adults (e.g., *Archer & Garcia, 2014*). Interestingly, in a sample of Swedish adolescents, locomotion predicted frequently exercising, which in turn predicted higher grades, while assessment predicted higher grades but also lower levels of well-being (*Garcia et al., 2015c*). These findings suggest that interventions need to be holistic or biopsychosocial, that is, targeting all aspects of self-regulatory mode in order to target both academic achievement and well-being (cf. *Garcia et al., 2016a*). We suggest that the RMQ could be used as a tool in this endeavor. In addition, since regulatory mode can be situationally induced (*Avnet & Higgins, 2003*) and situations may influence the preference of one mode over the other (*Kruglanski et al., 2010*), then, educators and other practitioners might design person-centered interventions that enhance or promote locomotion orientation through movement or exercise.

In spite of its theoretical and practical relevance, the present study has some limitations. We didn't include other measures of individual differences (e.g., Big Five) in order to test the incremental validity of regulatory mode in the prediction of affective well-being. Even if previous studies have shown that the relationship between regulatory mode and, for example, psychological vitality or energy, remain unchanged while controlling for the Big Five personality factors (*Kruglanski et al., 2000*), it would be useful to address this issue in cross-cultural studies (cf. *Delle Fave & Bassi, 2009*, who suggest cross-cultural studies as important to disentangle happiness and related constructs). It is also worth noticing that our sample was unbalanced with respect to gender. The higher presence of males, compared to females, could lead to a selection bias. Despite that earlier studies, using samples from other cultures, have not indicated any significant gender differences (*Kruglanski et al., 2000*), it could be interesting to investigate the presence/absence of this difference in a Swedish sample and its effect on the factor structure of the RMQ. After all, gender differences seem

to depend on the level of egalitarianism within the culture—although counter-intuitive, gender differences in, for example, personality traits, self-esteem, subjective well-being, and depression are larger in relatively high gender egalitarian cultures, such as Sweden (*Schmitt et al., 2016*). Another concern is that our sample only comprised high school and university students. Hence, further studies should include a more heterogeneous sample to address differences in, for example, gender, education level and also age. Moreover, although most of the studies using the RMQ disregard of the lie scale, as we did, and some have also applied only one of the scales (e.g., locomotion; *Pierro et al., 2013*), it might be useful to consider the presence of social desirability in future studies. Also, longitudinal designs would provide insights on the Swedish RMQ's test-retest validity and perhaps also a better understanding of the causal relationships between regulatory mode and affectivity.

Finally, we urge person-oriented approaches for the investigation and development of Regulatory Mode Theory, especially since peoples' tendency to prefer either locomotion and assessment is probably influenced by unrelated biological (e.g., temperament), psychological and social resources and conditions (*Kruglanski et al., 2000*). In other words, locomotion and assessment are independent and orthogonal biopsychosocial modes of operation, thus, some individuals can be high on one mode and low on the other mode, low on both modes, or high on both modes (cf. *Garcia & Lindskär, 2016*). Person-oriented statistical approaches are methods that focus on internal patterns, in contrast to variable-oriented approaches which focus on individual differences (*Lundh, 2015*). For instance, from a person-oriented framework these two modes of operation (locomotion and assessment) within the individual can be seen as interwoven components with whole-system properties (*Bergman & Wångby, 2014*). The outlook of the individual as a whole-system unit is then best studied by analyzing patterns of information (*Bergman & Wångby, 2014*; *Bergman, Magnusson & El-Khouri, 2003*; *Bergman & Wångby, 2014*; *Bergman & Magnusson, 1997*; see also *Cloninger, Svrakic & Svrakic, 1997*, who explain nonlinear dynamics in complex adaptive systems). There is, indeed, an increasing amount of person-oriented methods that can be used (*Garcia, in press-a*; *Garcia, in press-b*; *Garcia & Lindskär, 2016*; *Garcia, MacDonald & Archer, 2015*; *Bergman & Lundh, 2015*; *Valsiner, 2015*; *Molenaar, 2015*; *Laursen, 2015*; *Asendorpf, 2015*; *Von Eye & Wiedermann, 2015*; *Aunola et al., 2015*; *Baker, 2015*; *MacDonald & Kormi-Nouri, 2013*) as well as statistical software aimed for these endeavors (ROPstat; *Vargha, Torma & Bergman, 2015*).

## Conclusion and final remarks

One way to improve the applicability—and consequently the usefulness—of a measure is to verify its reliability and validity in different cultural contexts. The present study tested and confirmed the one-dimensionality structure and the reliability of the two scales of the RMQ (i.e., locomotion and assessment) in the Swedish context. Moreover, the present study confirmed the discriminant validity of self-regulation modes on affective well-being dimensions. This is important because the two affectivity dimensions are also measures of anxiety, which involves high negative affect, and depression, which involves both high negative affect and low positive affect (*Clark & Watson, 1991*). We suggest that the Swedish version of the RMQ, with only minor modifications, is a useful instrument to tap individual

differences in locomotion and assessment. In sum, the present study contributes to the validation of the RMQ in the Swedish culture, adds support to the theoretical framework of self-regulatory mode, and suggests that the regulatory mode system could be a psychological complex adaptive system, that along with the biological affectivity system, forms a larger biopsychosocial system of self-regulation.

### Funding

The development of this article was funded by a grant from the Swedish Research Council (Dnr. 2015-01229). The funders had no role in study design, data collection and analysis, decision to publish, or preparation of the manuscript.

### Grant Disclosures

The following grant information was disclosed by the authors:
Swedish Research Council: Dnr. 2015-01229.

### Competing Interests

Dr. Danilo Garcia is the Head of Research of the Blekinge Center of Competence, the Blekinge County Council's research and development unit. The Center works on innovations in public health and practice through interdisciplinary scientific research, person-centered methods, community projects, and the dissemination of knowledge in order to increase the quality of life of the habitants of the county of Blekinge, Sweden. He is also an Associate Professor at the University of Gothenburg and together with Professor Trevor Archer and Associate Professor Max Rapp Ricciardi, the leading researcher of the Network for Empowerment and Well-Being. This is an international network of junior and senior researchers and students who are interested in the field of well-being research.

Clara Amato is a senior researcher (FoU-ledare) at the Blekinge Center of Competence and a member of the Network for Empowerment and Well-Being.

Ali Al Nima is the main statistician at the Blekinge Center of Competence and a member of the Network for Empowerment and Well-Being.

Marko Mihailovic is a member of the Network for Empowerment and Well-Being.

### Author Contributions

- Clara Amato and Marko Mihailovic wrote the paper, reviewed drafts of the paper.
- Ali Al Nima conceived and designed the experiments, performed the experiments, analyzed the data, wrote the paper, prepared figures and/or tables, reviewed drafts of the paper.
- Danilo Garcia conceived and designed the experiments, performed the experiments, contributed reagents/materials/analysis tools, wrote the paper, prepared figures and/or tables, reviewed drafts of the paper.

### Human Ethics

The following information was supplied relating to ethical approvals (i.e., approving body and any reference numbers):

After consulting with the Network for Empowerment and Well-Being's Review Board we arrived at the conclusion that the design of the present study (e.g., all participants' data were anonymous and will not be used for commercial or other non-scientific purposes) required only informed consent from the participants.

## Data Availability

The raw data has been supplied as Data S1.

## Supplemental Information

Supplemental information for this article can be found online at http://dx.doi.org/10.7717/peerj.4092#supplemental-information.

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
