# Peer review of "Modus operandi and affect in Sweden: the Swedish version of the Regulatory Mode Questionnaire"

_PeerJ, doi:10.7717/peerj.4092_

## Round 0.1 · original submission · Major Revisions

We thank you for your submission of this manuscript at PeerJ. The three reviewers are positive about the strengths of the manuscript, but have also identified a number of more major amendments which should be attended to prior to resubmission. Of particular note, they have identified aspects of the introduction, methodology, results and discussion contain several comments which require more information as needing improvement. I would therefore recommend you carefully read these comments and consider how you can improve the manuscript to the standard required by the three reviewers.

Thank you for considering PeerJ for publication of your manuscript.
Best Wishes,

Reviewer 1 ·

Basic reporting

No comment.

Experimental design

No comment.

Validity of the findings

No comment.

Additional comments

The article is well written, clear and respects the professional standards of structure, introduction and background, literature references, ethical issues. It is an article well-founded theoretically, with a clear and precise objective. It was intended to verify the independence of the two regulatory modes and the psychometric properties of RMQ in the Swedish context. These objectives have been achieved in an appropriate manner. No flaws were found in the experimental design, although the student-only sample could be improved. However, for the validation of the scale is a correct sample. The discussion section is very interesting and complete. The study will serve to improve the literature on this topic.The section Limitations and suggestions for future research is very extensive, even in some moments the wording is unclear. I suggest reviewing this section.

Reviewer 2 ·

Basic reporting

This manuscript investigated two regulatory modes using structural equation modeling. The manuscript is well-organized. However, the statistical analyses need improvement.

Experimental design

Structural equation models presented on Page 14 need to be further specified before any analysis. The model specification is usually guided by a combination of theory and empirical results from previous research.

Validity of the findings

It would be helpful to present a table of parameter estimates (i.e. standard deviation, test statistics) and equation models for the SEM.

Additional comments

Summary

This manuscript investigated two regulatory modes using structural equation modeling. The manuscript is well-organized. However, the statistical analyses need improvement.

Major comments
1. Preliminary analysis
In the abstract, there were 5 participants did not report their gender. However, the percentage of male is much higher than female. That difference may lead to selection bias. Can the authors explain the data? In addition, preliminary descriptive statistical analysis such as collinearity issue and outlier detection need to be conducted carefully.

2. Model
Structural equation models presented on Page 14 need to be further specified before any analysis. The model specification is usually guided by a combination of theory and empirical results from previous research.

3. Results
It would be helpful to present a table of parameter estimates (i.e. standard deviation, test statistics) and equation models for the SEM.

Reviewer 3 ·

Basic reporting

No comment.

Experimental design

It seems that the analysis combines the university and high school students. I did not get any good sense as to why this was done. It is not clear whether separate analysis should be done for these two groups of students or not. Some clarification is needed.

Validity of the findings

No comment.

Additional comments

1. Some sentences are repeated word by word: the sentence ``However, other values (the goodness of fit index and the root mean square error of approximation) indicated that the hypothesized model fits the data well.’’ appears in line 235-236 and again in lines 253-255.

2. Related to the previous point, what are these values (the goodness of fit index and the root mean square error of approximation)? Please report the numbers.

---

## Round 0.2 · accepted · Accept

We thank you for attending to the reviewers comments on your revised
manuscript and am happy to let you know the paper has now been
accepted for publication

Reviewer 1 ·

Basic reporting

No comment

Experimental design

No comment

Validity of the findings

No comment

Additional comments

The authors have taken into account the suggestions and recommendations that the reviewers have indicated in their reports. The article may be published.